# Fostering Policy Change in Anti-Poverty Schemes in Italy: Still a Long Way to Go

**Franca Maino \*** and **Celestina Valeria De Tommaso \***

Department of Social and Political Sciences, University of Milan, 20122 Milan, Italy
\* Correspondence: franca.maino@unimi.it (F.M.); celestina.detommaso@unimi.it (C.V.D.T.)

**Abstract:** This article explores the poverty phenomenon and anti-poverty policies in Italy, before and after the COVID-19 pandemic. It aims to contribute to the mainstream literature on policy change, looking at how the convergence of multiple streams (problem, policy, and political) contributed to achieving the adoption of the Italian Minimum Income scheme, the Citizenship Income. Despite increasing political and public awareness of poverty, the 2022 Budget Law failed to achieve a structural reform—considering amendments proposed by the Ministry of Social Policy's Commission and the Italian Anti-Poverty Network—to improve both the equity and efficiency of the anti-poverty measure. Strong path dependency in the conceptualization and implementation of the anti-poverty tool is still evident; policy change thus has a long way to go.

**Keywords:** minimum income; poverty; policy change; COVID-19 pandemic

## 1. Policy Change and Window(s) of Opportunity

The "window of opportunity" has been used to describe a fertile scenario for the uptake of an idea, translated into a (new) policy initiative. Many juxtaposed elements may explain the opening of a window of opportunity. Existing research (Rose et al. 2018) suggests that these windows open when the chances of evidence-informed policy are improved (e.g., via technical information and reports). According to Kingdon, who proposed the idea of "window of opportunity", the policy process may be unpredictable and serendipitous. There are, however, various strategies that can be used to predict and facilitate the opening of windows for policy change (Kingdon 1984). Three resources contribute to understanding, assessing, and opening policy windows. In the field of the multiple stream approach (Kingdon 1984), the likelihood of successful agenda setting increases if all three elements—problem, proposal, and politics—are linked in a single package. What issues are included in the agenda or not in the public policy-making process is thus determined by participants inside and outside governments and the process, which includes the problem stream, the policy stream, and the political stream.

First, the problem stream relates to the source of the issue that necessitates the introduction of a policy. For example, increasing awareness of a policy problem (e.g., poverty) gaining ground at the national level, i.e., in the media and public opinion. Second, the policy stream involves evaluations and analysis of a wide range of policy ideas and proposals, directed to the solution of the problem. It comprises policy and business strategies to achieve a goal or a specific outcome. Third, the political stream includes a multitude of elements: government legislation, the influence of non-government organizations and pressure groups, and the influence of supranational actors (e.g., the European Union).

Each of these streams has their own distinct life; when they come together, a problem becomes relevant on the agenda, policies that match the problem receive attention, and a fertile ground for policy change becomes possible. The window of opportunity, when opened, allows for the streams to interact (Crank 2003). However, the juxtaposition of these streams is not predictable: policy entrepreneurs[1] advocate their own proposals to



gain the chance to achieve policy change (Cairney 2011). Their roles have been described by Kingdon (1984) as actors who use their knowledge of the process to further their own policy ends, taming the "unpredictability" in policy change. According to this scenario, the multiple stream model indicates that the successful proposal and implementation of a policy depends on the level of integration of all streams discussed above, also taking into account the policy entrepreneurship role to allow better convergence of these three streams.

This model fits the proposed case study: anti-poverty policy measures in Italy. With reference to our case of analysis, the convergence between the policy, problem, and political stream reached its climax in 2019 when the national basic income—the Citizenship Income—was introduced. The historical path leading to the achievement of the Italian anti-poverty measure proves the gradual confluence of increasing awareness of poverty, as a multi-faceted phenomenon, the structuring of political strategies to alleviate the phenomenon, and coordination among political and non-political actors, also involving an advocacy coalition, the Italian Anti-Poverty Network (l'Alleanza contro la Povertà).

The introduction of an anti-poverty measure showed its positive effects during the COVID-19 pandemic: it managed to alleviate the socio-economic consequences due to the emergency period. However, it was not enough to cope with the increasing numbers of individuals and families in absolute poverty, a large part of whom are not entitled to receive the Citizenship Income. In fact, in 2020, an extraordinary and temporary anti-poverty measure—the Emergency Income (Reddito di Emergenza, REM)—was introduced to cope with the increase in relative and absolute poverty. Why does the adoption of an Emergency Income constitute a relevant element for the analysis of policy change in poverty reduction? The introduction of a temporary measure proved path dependency and policy legacy in Italian social assistance and anti-poverty policies, characterized by a fragmentary and categorical approach (Madama 2010). The criticalities of Citizenship Income—notably, RdC is a means-test measure whose structural requirements tend to penalize short-term resident immigrants and large families; moreover, it fails to achieve the goal of beneficiaries' employment and re-employment in the labor market, thus being classified among mere in-cash transfers rather than in-kind benefits—unveiled how a paradigmatic shift towards universalism in fighting poverty is still far from occurring. At the end of 2021, the Emergency Income was cancelled, and the 2022 Budget Law failed to accept the amendments to the RdC proposed by the 2021 Ministry of Social Policy's Commission and by the Italian Anti-Poverty Network (l'Alleanza contro la Povertà) to guarantee a more comprehensive coverage through anti-poverty policy.

The article is structured as follows[2]. Firstly, the research question and methodology are described (Section 2). The third section introduces the Citizenship and Emergency Income policy design (Section 3), including the main historical milestones that framed the introduction of the two measures. Secondly, the article analyzes two ineffectiveness dimensions related to anti-poverty policies (Section 4): the increasing poverty rate in Italy and its take-up rate, including the characteristics of the distribution among beneficiaries. The fifth section concludes.

## 2. Research Questions, Hypothesis, and Methodology

The interaction among the problem, policy, and political streams led to Citizenship Income's adoption in 2019. However, the window of opportunity to ameliorate the measure and overcome its criticalities closed: despite increasing pressures on the welfare state, due to the COVID-19 crisis, the Emergency Income was introduced in 2020 to enhance the capacity of welfare provisions to cope with the increasing poverty rate and was abolished 1 year later in 2021. Following, the Budget Law 2022 did not introduce any structural change to anti-poverty policies in Italy.

For enhanced clarity, the following research question is contextualized in the Italian welfare state. The latter has traditionally presented a distributive and functional imbalance (Ferrera 2012). The first one refers to the imbalance of social public expenditures towards old-age and pensions. The distributive distortion instead concerns imbalances in resources

in favor of the *insiders* employed (those with a fixed and standardized employment contract). These are deemed to be sensitive issues in the public debate on welfare reform in Italy. Moreover, in the last decades, this implied limited resources for families with children and virtually non-existent anti-poverty programs (Ferrera 2012).

This paper thus aims to explore the following research question: "What are the main dimensions of the ineffectiveness of the Citizenship Income? What are the rationales behind the failure of anti-poverty reform in 2022?". The hypothesis states that poverty and growing inequality are still a divisive theme among political and non-political actors. They polarize around the main functions of minimum income schemes: an instrument as a safeguard against low income and poverty (Immervoll 2009) or as an instrument for job activation and re-employment (Marchal and Mechelen 2013). An intervening factor to explain the above-mentioned hypothesis concerns the COVID-19 pandemic and the saliency of health-related issues. Given this polarization, the second half of 2021—focused on the third pandemic wave and mass vaccination procedures—left no room for the converging of different positions towards policy solutions to the criticalities of the Citizenship Income. This may have contributed to lower relevance of and urgency regarding the anti-poverty scheme on the policy agenda.

Through a descriptive and analytical approach, the aim of this paper is to piece together the historical milestones that led to the national anti-poverty measure in 2019, emphasizing the main criticalities of the measure in tackling the increasing poverty rate at the national level. This research was conducted using qualitative methods: content and desk analysis. As stated in footnote 1, this paper is part of a research project that the authors conducted in the field of poverty, as a multidimensional phenomenon, and has, therefore, benefited from original data collected through a focus group[3].

### 3. Two Italian Measures to Combat Poverty: Citizenship Income and Emergency Income

This section aims to present the historical milestones towards the introduction of a minimum income scheme in Italy: Citizenship Income. The first part introduces the Italian path to adopting a guaranteed minimum income (Section 3.1). The second and third parts present the Citizenship Income and the Emergency Income (Sections 3.2 and 3.3), retracing their mode of function.

#### 3.1. The Italian Path towards a Guaranteed Minimum Income

In 2019, the adoption of the Citizenship Income aligned Italy with other European countries, introducing a means-tested guaranteed minimum income. The Italian path[4] towards a national measure to combat poverty gained ground in 2008 when the Berlusconi government introduced the "Carta Acquisti": a debit card with 40 euros per month, aimed mainly at retired people to facilitate the acquisition of primary goods. The policy measure was then inherited by the Monti government, who, in 2013, launched the "New" Carta Acquisti to low-income households with minors (at least one). The policy required the beneficiaries to sign a "Social Inclusion Path"[5]. The latter policy was mainly managed by local municipal services.

This policy measure was firstly piloted for a 12-month period. The pilot started in 2013 and reached less than 10 percent of eligible households. The eligibility criteria were exceptionally stringent. Among others, households with a minor had to hold an Equivalent Economic Situation Indicator (ISEE)—a means-test that allows for evaluation of the economic situation of families by considering the income, assets, and composition of the families jointly—lower or equal to 3000 euros, and could not own a house with a value exceeding 30,000 euros nor a movable asset exceeding 8000 euros. Moreover, working-age household members had to demonstrate that they had stopped working at least 36 months before the application to receive the in-cash transfer.

The experiment was conducted in 11 cities in Italy. The measure failed to achieve the desired outcomes: only a small amount of money was efficiently used by local governments to reach potential beneficiaries. In 2013, the Letta government set up an inter-ministerial work-

ing group to discuss the policy toolbox to design an Italian minimum income. The group presented a proposal to introduce Support for Active Inclusion (Sostegno all'Inclusione Attiva, SIA), a universalistic policy measure that considered lack of income as the sole eligibility criterion. Additionally, in this case, the provision of the subsidy was accompanied by an "Inclusion Path"[6], thus confirming the orientation towards active labor market inclusion alongside poverty alleviation.

In 2013, the Alleanza contro la povertà (the "Anti-Poverty Network") was founded to bring together a vast number of social actors to contribute collectively to the construction of adequate public policies to prevent absolute poverty in Italy. The Anti-Poverty Network, still operational, was made up of 36 organizations: associations, representatives of municipalities and regions, the Third Sector, and trade unions. In 2013, to structure its political confrontation, the network developed and presented its own policy proposals, whose effects were suddenly evident in subsequent policy decisions. Nonetheless, the Stability Law 2014 (i.e., Budget Law) did not allocate a sufficient amount of resources needed to implement the policy. It followed a complex recovery process led by the government to gather a multitude of financial resources to allow the extension of the Nuova Carta Acquisti, now renamed "Support for Active Inclusion". The policy measure has a weaker intervention capacity than the one expected by the inter-ministerial working group; however, it was a strong political signal to carry on the policymaking process towards a national (therefore, universalistic) minimum income.

In 2015, the Renzi Government intervened in an incisive way. First, the adoption of a Delegated Law was envisaged for the promotion of a "single national measure to combat poverty". Secondly, the Fund for the Fight against Poverty and Social Exclusion was established. For the first time, structural funding (i.e., permanently registered in the public finance register) was provided for the fight against poverty. In particular, 380 million euros were allocated, together with the resources recovered from previous allocations, and were committed to further extending the SIA. The SIA became a "bridge measure" (pending the "Reddito di Inclusione", ReI) aimed at supporting families in poverty throughout the national territory. The Stability Law 2017 finally approved the allocation of 1 billion euros per year when fully operational. The Anti-Poverty Network had a strong influence in achieving the first step towards the Support for Active Inclusion and Inclusion Income, first, and Citizenship Income later (Jessoula and Madama 2015; Gori 2020).

Between 2016 and 2017, the Support for Active Inclusion was implemented and extended to the whole country. It relaxed the eligibility criteria for potential beneficiaries, among individuals and families (Agostini 2014). Despite this, many people were still excluded from receiving benefits. In 2018, the Gentiloni government established the Income for Inclusion (ReI). Before even entering into force, the Budget Law 2018 transformed the ReI into a universalistic scheme. The eligibility criteria were restricted to some conditions, mainly referring to citizenship and the economic situation. The Poverty Fund was increased by 300 million euros in 2018, 700 million euros in 2019, and 900 million euros in 2020. Therefore, the measure's take-up was extended. The Gentiloni Government recognized protection from poverty as a subjective right (Agostini 2014; Gallo and Luppi 2019; Gori 2020): the government introduced a national policy measure to combat poverty identified as an Essential Level of Benefits (the so-called, "Livelli essenziali delle prestazioni", LEP).

*3.2. The Citizenship Income*

In 2019, by Law Decree no. 4/2019, the Citizenship Income was introduced and replaced the ReI. The legislation defines the RdC as a measure of *"active labour policy to guarantee the right to work, to combat poverty, inequality and social exclusion ( . . . )"*. When the beneficiaries are persons aged 67 or over, the RdC is called the Citizenship Pension (PdC). In line with the previous measures, the RdC consists of an in-cash transfer provided monthly through an electronic card (the RdC Card). Although its name suggests a universal and unconditional basic income, the RdC is a means-tested cash benefit targeted at poor and socially excluded households and is conditional on participation in job-search activities.

Compared to the ReI, the RdC is supported by more public financial resources, it is more inclusive, and has an 18-month duration, so longer than ReI (for a comparison between ReI and RdC, see Jessoula et al. 2018).

The measure's eligibility criteria require households to have a maximum annual ISEE (indicator of equivalized economic conditions, considering both income and wealth) of 9360 euros and an annual equivalized income no higher than 6000 euros. Moreover, housing (excluding primary residence) and financial wealth must not exceed 30,000 euros and 10,000 euros, respectively. The household equivalent income is measured on an equivalent scale. The scale is equal to 1 for the first member of the family unit. It is increased by 0.4 for each additional member over the age of 18 and by 0.2 for each additional minor member, up to a maximum of 2, raised to 2.2 in the presence of household members with a severe disability or non-self-sufficiency. Concerning active participation in the labor market, one of the eligibility criteria is compliance with a "personalized" path towards social inclusion and job placement. The local social services are required to carry out a multifaceted assessment of the family household needs.

The RdC introduces two main innovations. Firstly, for the first time, a measure of selective universalism in the fight against poverty is introduced structurally with the aim of achieving a homogeneous essential level of benefits all over the country; it involves a complex set of social inclusion and active labor market policies and measures managed by very different local actors, so posing a problem of multilevel governance and institutional networks (i.e., coordination, cooperation, and integration in providing services) that should work homogeneously all over the country. Secondly, in a context in which the fight against poverty has traditionally played a residual role, the introduction of these measures moves toward a recalibration of the welfare system by intervening in the functional distortion[7] (Ferrera 2012). Most public resources are traditionally subject to some risks (i.e., primarily old age, through a hypertrophic pension system). In contrast, other threats—including poverty—are little considered, receiving few public resources through rudimentary programs. Secondly, the ReI and RdC end the era of "policy experimentation" in combating poverty (Matsaganis et al. 2003). The RdC has marked a paradigmatic change.

*3.3. The Emergency Income*

The Emergency Income was introduced in 2020. The measure was adopted on a temporary basis in 2020 to alleviate new poverty during the emergency period. The eligibility criteria required beneficiaries to hold an ISEE value lower than 15,000 euros and movable assets not exceeding 10,000 euros, so imposing weaker restrictive eligibility criteria than RdC. Unlike the RdC, the REM granted the full amount to all those who meet the requirements (Natili et al. 2021). The in-cash transfer was aimed at those not earning the Citizenship Income or other allowances introduced by the government to cope with the COVID-19 crisis. The contribution varied from a minimum of 400 euros per month to a maximum of 800 euros (which can be raised to 840 euros for the severely disabled or non-self-sufficient people). The initial allocation of public resources amounted to 971 million euros. In 2020, it managed to reach 425,000 recipients. The REM was renewed three times during 2020 and re-planned once during 2021[8].

The measure was renewed three times: twice during 2020 and once in 2021[9]. Contrarily to RdC, the REM did not require beneficiaries to follow an inclusion path, due to its transitory nature. The REM played the role of a "lifeboat" for those who were worse off: a monetary contribution aimed at families in severe economic hardship as a result of the spread of COVID-19.

## 4. Two Dimensions of Ineffectiveness: Equity and Efficiency

During the 2008 and 2014 economic crisis, Italy lacked a minimum income scheme. This had dramatic consequences on the poverty rate (Jessoula and Madama 2015; Gori 2020; Natili 2019). From 2018, it was possible to fill the gap: Italy introduced the first structural scheme of a national Minimum Income. After decades of scarce attention to the poverty

issue—and, even, policies to combat it—Italy radically transformed the income protection system, introducing the ReI and RdC.

However, the implementation of RdC has been characterized by at least two inefficiencies, in terms of equity and efficiency. Despite severe worsening of poverty conditions at the national level, the Budget Law 2022 did not manage to achieve a structural reform. The multiple stream convergence failed to foster policy change. What are the main ineffectiveness dimensions? Citizenship Income has been characterized by inefficiencies related to equity (the take-up rate) and efficiency (pathways to social and work inclusion).

The first dimension is explored through contextualization of "multidimensional poverty" in Italy (Section 4.1). The second dimension, efficiency, is explored by data analysis of Citizenship Income beneficiaries' rate, profiles, and geographical distribution (Section 4.2).

### 4.1. Poverty in Italy: Before and after the COVID-19 Crisis

In June 2022, the Italian Statistical Institute[10] published updated data about poverty in Italy. Three trends are worth mentioning for better understanding the poverty rate in the last two years (2020–2021). Firstly, in 2021, the incidence of absolute poverty[11] was 7.5% among households (from 7.7% in 2020) and 9.4% among individuals (same value as the previous year): more than 1.95 million households, for a total number of about 5.6 million individuals. Secondly, the year 2021 was still marked by the pandemic but with a strong economic recovery (+6.6% of GDP). Thirdly, household consumption expenditure, according to the results of the Household Budget Survey, increased (+4.7% in current terms compared to the previous year), without compensating the decline in 2020 and still showing a decrease of 4.7% compared to 2019.

Poverty stability may be explained by rapid economic growth, alongside a scarce increase in the expenditure rate for consumption among low-income families, together with a higher inflation rate[12] (+1.9% in 2021) (ISTAT 2022). In the last 16 years, the absolute poverty rate has increased by 4.1 percentage points in Italy (Figure 1). The highest rates are those in southern Italy (10% in 2021), followed by the north (6.7%) and the center (5.6%).

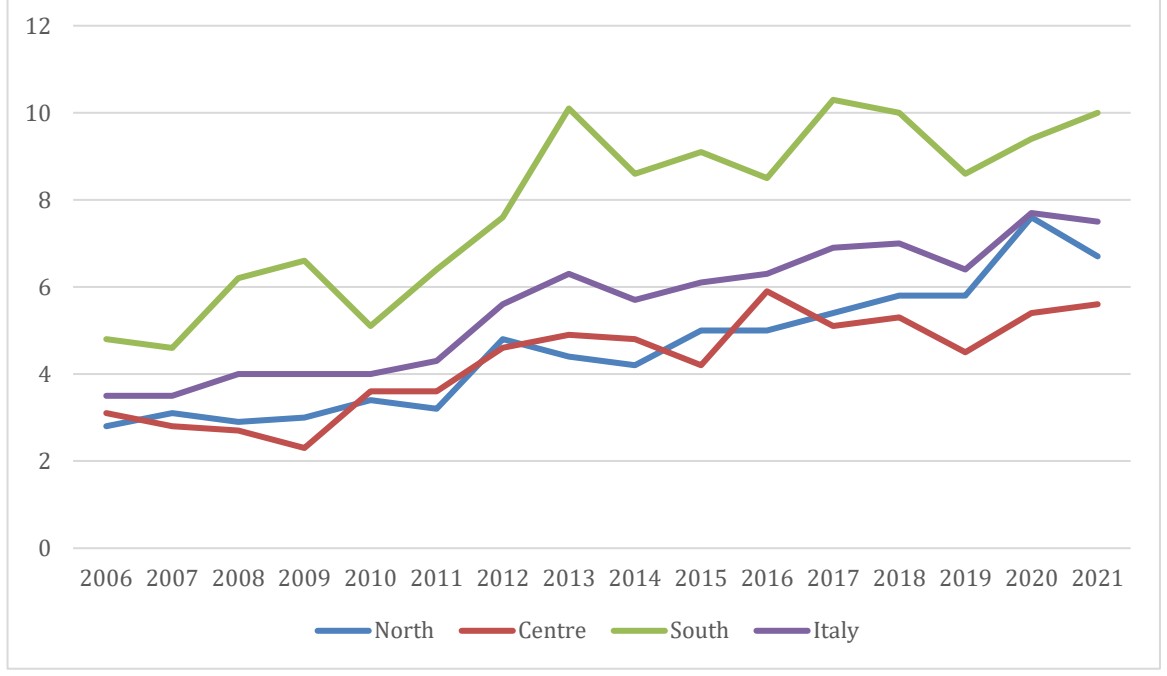

**Figure 1.** Poverty rate trends in Italy, 2005–2021. Source: own elaboration on the Istat database.

From 2020, the poverty rate decreased by 0.2 percentage points for individuals and kept the same percentage for households. The rate in southern Italy increased from 9.4%

to 10% for individuals and from 11.1% to 12.1% for households. The latter are the largest poverty values in Italy (Figure 2).

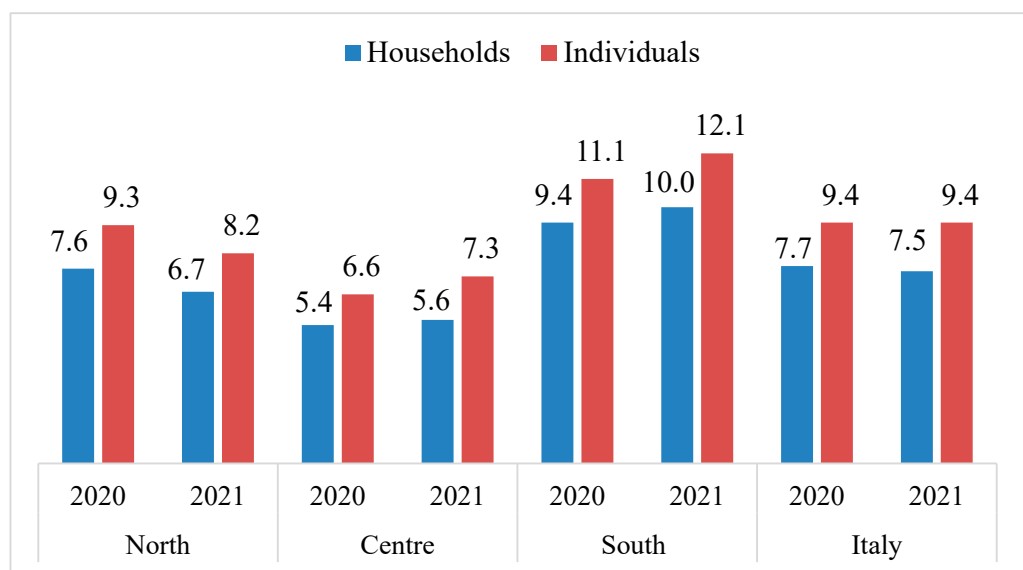

**Figure 2.** Poverty rate trends among households and individuals, 2020–2021. Source: own elaboration on the Istat database.

The poverty rate has been increasing for minors (0–17 years old), from 13.5% in 2020 to 14.2% in 2021. As with the other age cohorts, a slight growth (+0.2 percentage points) was also evident for those of a working age (18–64 years old) (Figure 3).

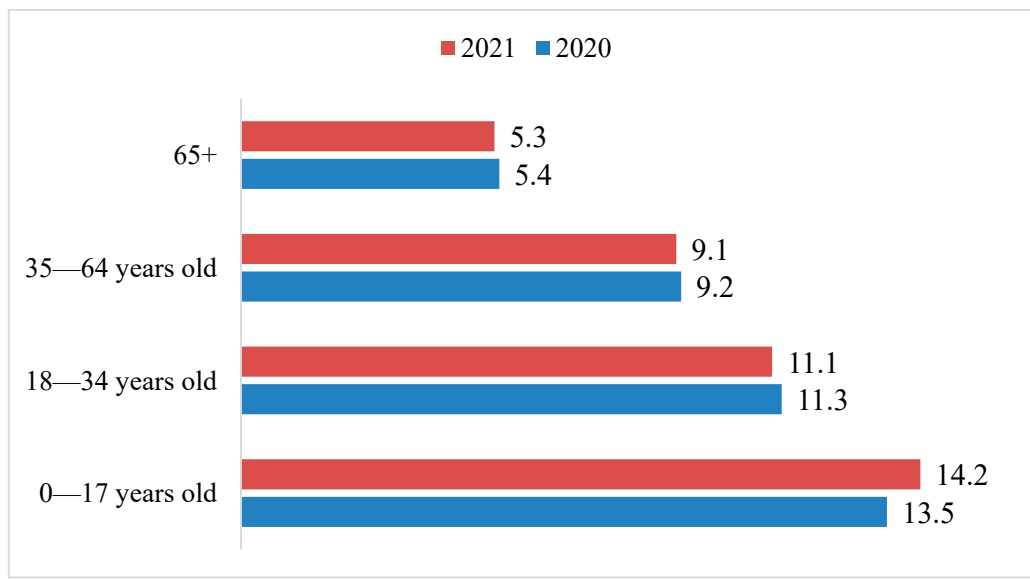

**Figure 3.** Poverty rate trends, by age, 2020–2021. Source: own elaboration on the Istat database.

Regarding households, the type of families more at risk of poverty are those with five members or more. Nonetheless, poverty has been increasing among single-individual families (from 5.7% to 6%) and those with four members (from 11.2% to 11.6%) (Figure 4).

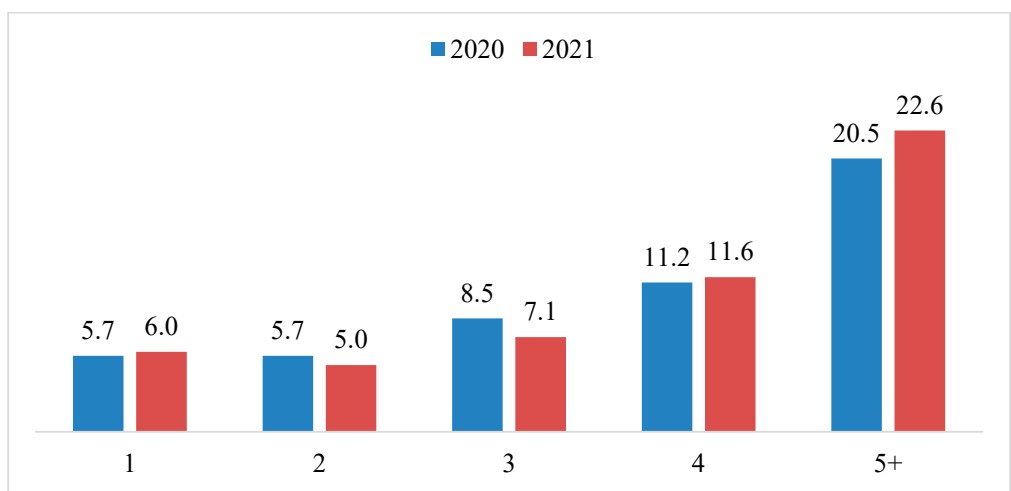

**Figure 4.** Poverty rate trends, by number of household members, 2020–2021. Source: own elaboration on the Istat database.

The greater the number of children, the higher the poverty rate among households. The percentage is equal to 22.8% among those with three children or more. However, it has also been increasing for households with two children (from 12.5% in 2020 to 14% in 2021). This value decreased for families with one child (from 9.3% in 2020 to 8.1% in 2021) (Figure 5).

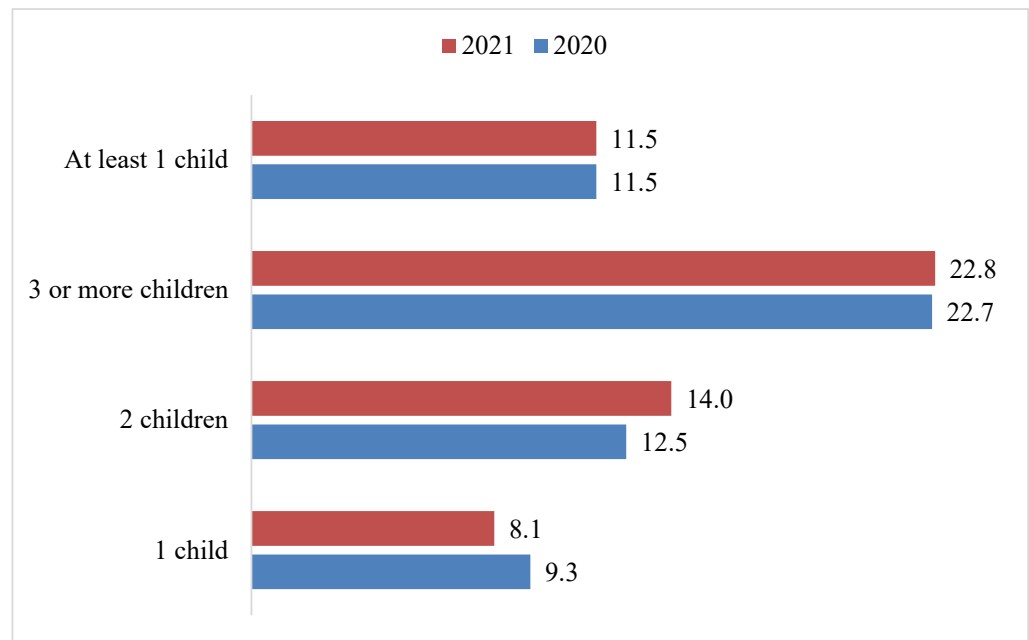

**Figure 5.** Poverty rate trends in households, by number of children, 2020–2021. Source: own elaboration on the Istat database.

As in the case of elderly, poverty is widespread among families with just one old-age relative (or at least one) in their household. Contrary to data shown in Figure 5, these trends continued in the last two years (Figure 6).

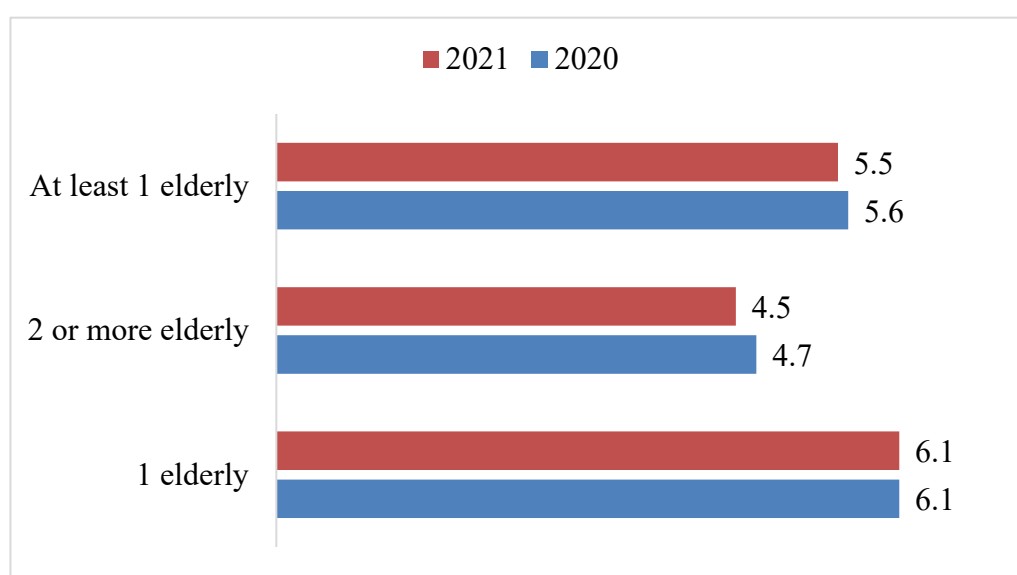

**Figure 6.** Poverty rate trends in households, by number of elderly people, 2020–2021. Source: own elaboration on the Istat database.

The risk of poverty is highest for foreign families. As shown in Figure 7, 30.6% of foreign families (5.7% of Italian ones) are at risk of poverty in Italy (+4 percentage points from 2020). The highest rate recorded is in southern Italy (37.6%) followed by the north (30.2%). All trends concerning foreign households have been increasing in the last two years, whereas a slight decrease occurred among Italian households (Figure 7).

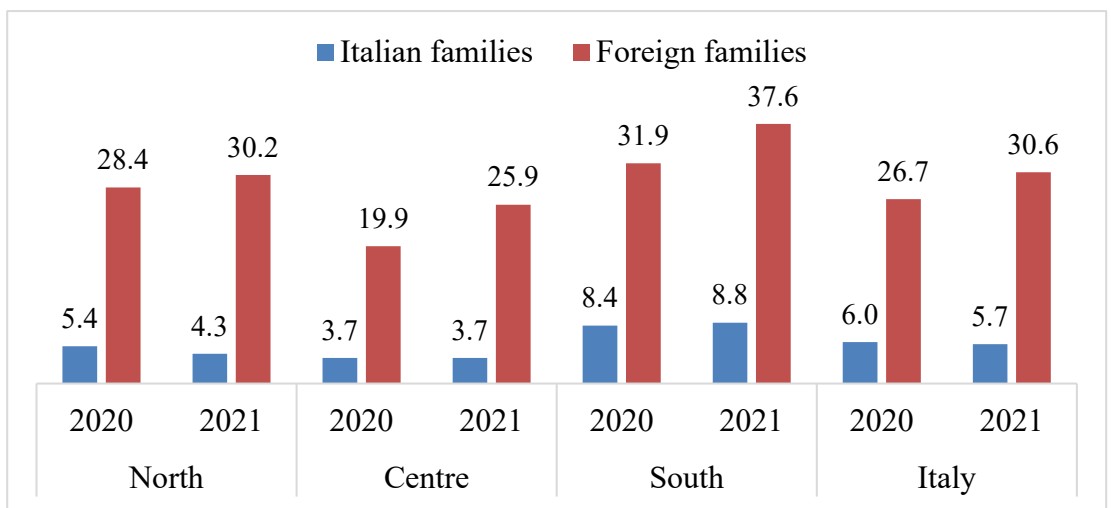

**Figure 7.** Poverty rate trends in Italian and foreign families, 2020–2021. Source: own elaboration on the Istat database.

To sum up, absolute poverty hit harder in the southern part of Italy, for large and one-person families, and families with children or elderly people. The highest rate of poverty is among foreign families, living in the north and south of Italy. Poverty rose among minors (0–17 years old) more than other age cohorts.

### 4.2. Poverty Is a Multidimensional Phenomenon

Few empirical studies have analyzed multidimensional poverty in Italy at a subnational level (Betti et al. 2008; Coromaldi and Zoli 2012; Giuliano et al. 2012). Despite little data, this section aims to develop the discussion about how several dimensions concur to explain increasing poverty and social exclusion, among adults and children.

Three dimensions are explored: in-work, educational, and food poverty. Other known dimensions—that are not considered here —concern housing, digital, and energy poverty. The proposed discussion is aimed at encouraging reflection about the relevance of in-kind services, alongside mere in-cash provisions, to empower social inclusion.

Having a job does not prevent falling into poverty. In 2020, 10.8% (8.9% in Europe) of Italian workers were poor (Eurostat database) (Figure 8). The rate is higher for men (10.8%)—due to their stronger participation in the labor market—than for women (8.5%). The younger the worker, the higher the risk of being poor despite working.

Related to this, the number of young NEET (not in employment, education, or training) reached the highest rate in Europe (24.4% in 2021 vs. 14.3% in Europe) (Figure 9). Mainstream literature and empirical evidence suggest that Italy's school-to-work transition is slow and difficult: the transition to work is difficult even for Italy's university graduates, who struggle to secure jobs and earn less than their European peers (Cigna 2020). Unemployment and the inactivity trap are thus exacerbated by educational and skill mismatch, with young overeducated workers with tertiary education or undereducated with secondary education at risk of unemployment (or, in the worst cases, inactivity).

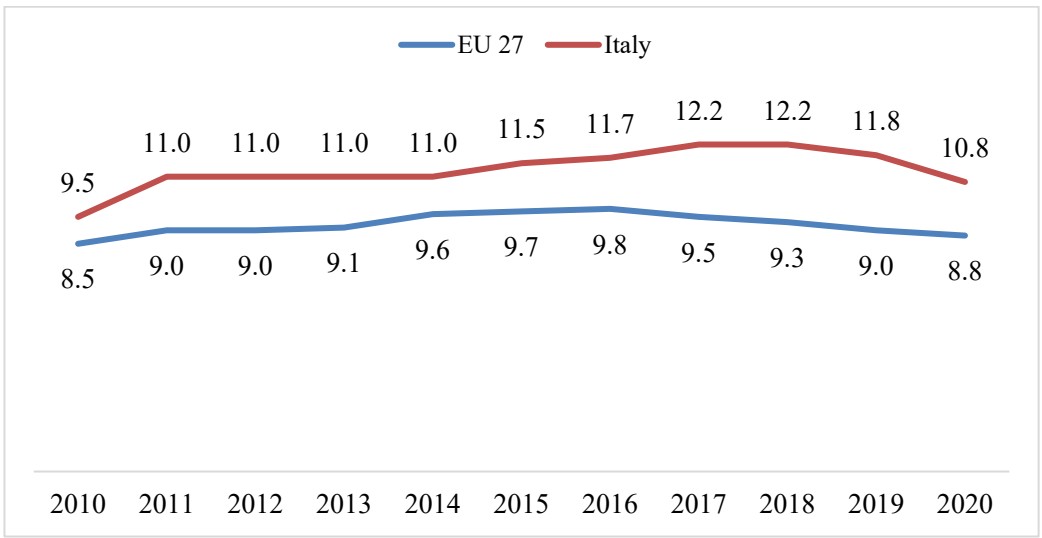

**Figure 8.** In-work poverty in Europe and Italy, 2008–2020. Source: own elaboration on the Eurostat database.

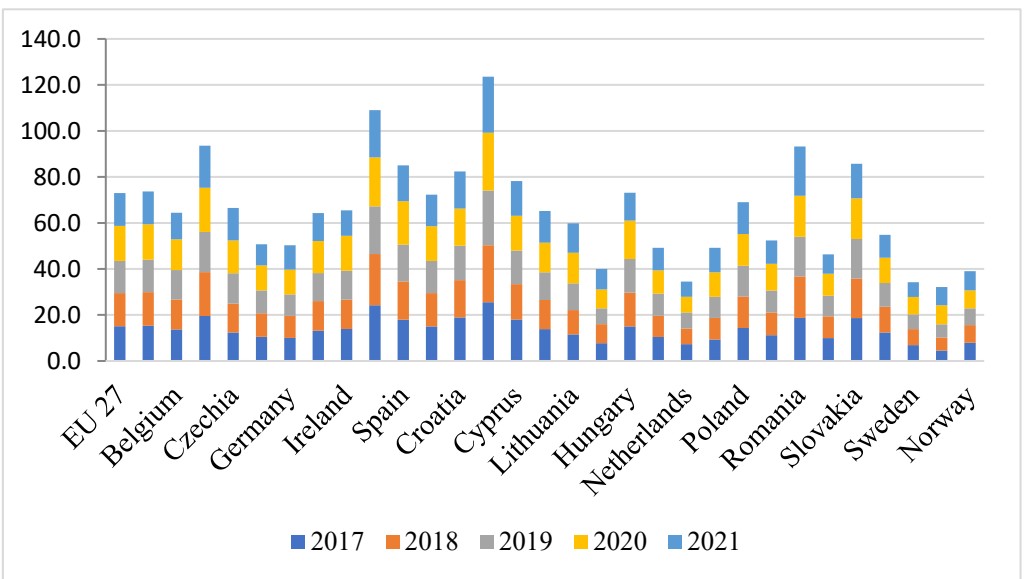

**Figure 9.** NEET in European countries, 2017–2021. Source: own elaboration on the Eurostat database.

Particularly noteworthy is that in 2021, Italy was the fourth European country[13] for early leavers from education and training among young people aged 18–24 years (12.7% in Italy vs. 9.7% in Europe). However, this percentage was 17% in 2012 (12% in Europe), experiencing a decrease in the last 10 years (Figure 10).

Alongside the deprivation of opportunities—in relation to educational poverty and slow transition to work as mentioned above—social exclusion is determined by high levels of material deprivation. The material deprivation rate is the EU-SILC indicator used to measure the inability to afford some items considered to be desirable (or necessary) to lead an adequate life. Among others is the inability to eat meat or proteins regularly. The inability to eat meat or proteins regularly was equal to 12.9% in 2020, 5 percentage points less than in 2016. The value is, however, one of the highest in Europe (Figure 11). This rate is equal to 5.5% in Italy (7.9% in Europe) among adolescents aged 12–17 years old and 5.7% (7.2% in Europe) among those aged 15–19. The wide-reaching impacts of food poverty showed how it might have a negative effect on the home environment, with parents often skipping meals to protect children and experiencing levels of stress: this is also associated with adverse effects on children's education and poor concentration at school, worse attendance and learning outcomes, and stigma and bullying (Actionaid 2021; Unicef 2022).

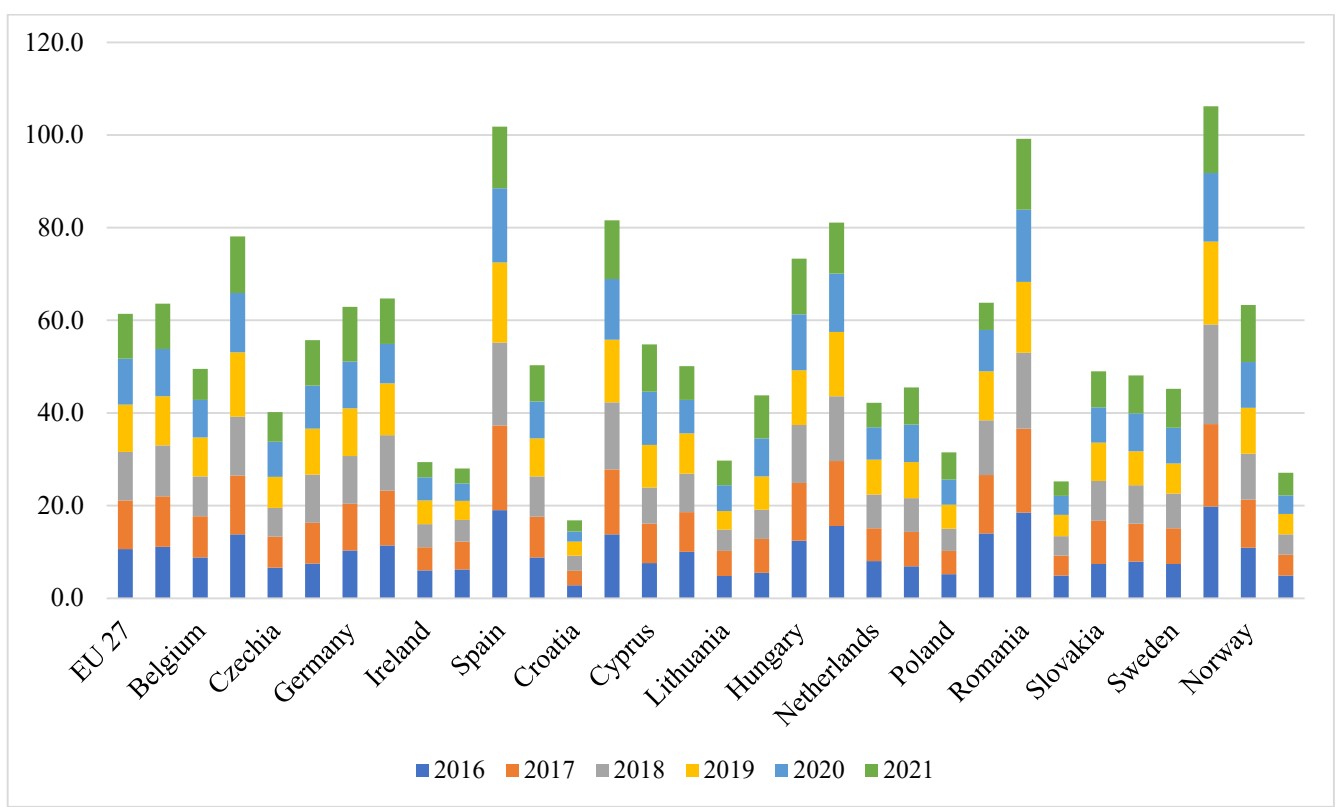

**Figure 10.** Early leavers from education and training in Europe, 2016–2021. Source: own elaboration on the Eurostat database.

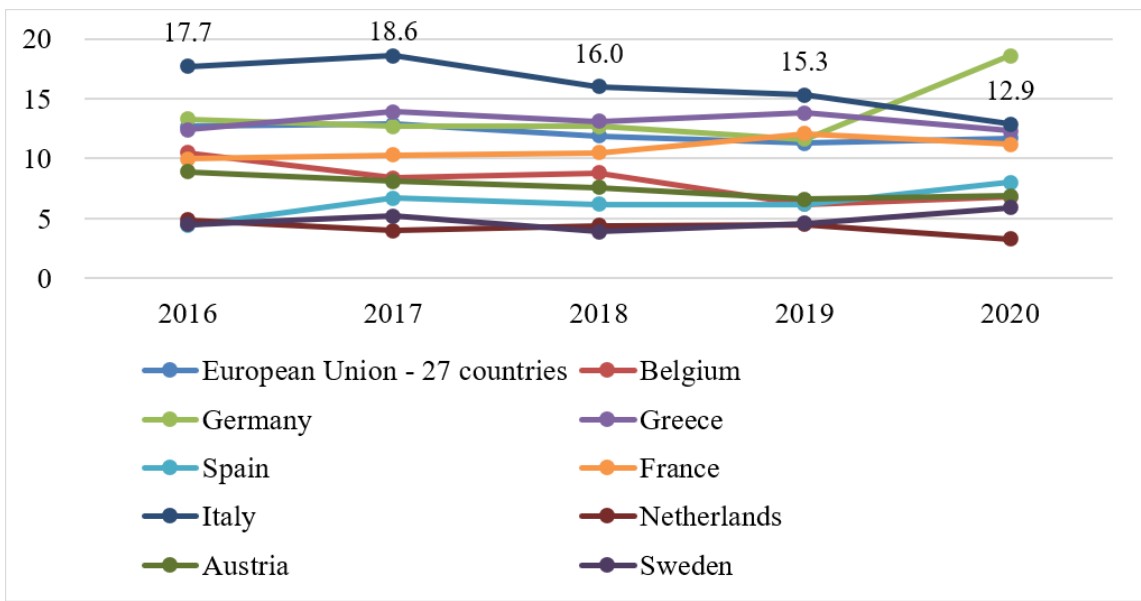

**Figure 11.** The inability to eat meat or proteins regularly in Europe, 2016–2021. Source: own elaboration on the Eurostat database.

Beyond monetary poverty, the phenomenon is enriched by a multitude of dimensions, which manage to grasp its multifaceted complexity. In Italy, the development of anti-poverty policies has been hindered by strong public social spending that is unbalanced towards old-age pensions and "insiders" in the labor market (i.e., employed persons with stable and full contracts). The welfare provisions are mainly linked to means-tested

criteria, supporting a workfare approach to social welfare policies. Nonetheless, increasing awareness of the poverty phenomenon led, in 2019, to the adoption of a Citizenship Income. As described above, awareness in political and public debate was not the only factor: an advocacy coalition, the Italian Anti-Poverty Network, contributed to achieving the desired outcome. The design of the Italian minimum income, however, did not allow all at-risk-of-poverty and poor people in the country to be reached. The Citizenship Income contributes to alleviating poverty at the national level; in the meantime, it proves inefficiencies related to equity (inability to reach the highest portion of poor people) and efficiency (mostly related to bottom-up implementation at the local level). As said, the sharply worsening poverty conditions pushed public actors to adopt a temporary measure, the Emergency Income (Busilacchi et al. 2021). The next section (Section 4.3) is thus aimed at grasping the equity and efficiency gaps in Citizenship Income implementation. The following data must be read considering widening inequality and increasing poverty during the COVID-19 era, as presented above.

### 4.3. The Citizenship Income: Who Is Targeted and What Are Its Characteristics?

The National Social Insurance Agency (INPS) provides a monthly release of data on Citizenship Income (and Pensions)[14]. The Observatory on Citizenship and Pension Income reports data about the number, profiles, and characteristics of beneficiaries in terms of the geographical distribution, monthly average amount, and household composition. In 2019, the number of welfare recipients of PdC were 1.1 million, for a total of 2.7 million people involved. This quota increased in 2021: the recipients of at least 1 month's provision were almost 1.8 million, with 4 million people involved. As of 2022, the highest rate of RdC household beneficiaries is in Campania (19.8% of total national beneficiaries), Sicily (17.6%), Puglia (9.4%), Lazio (8.7%), and Lombardia (8.2%) (Figure 12).

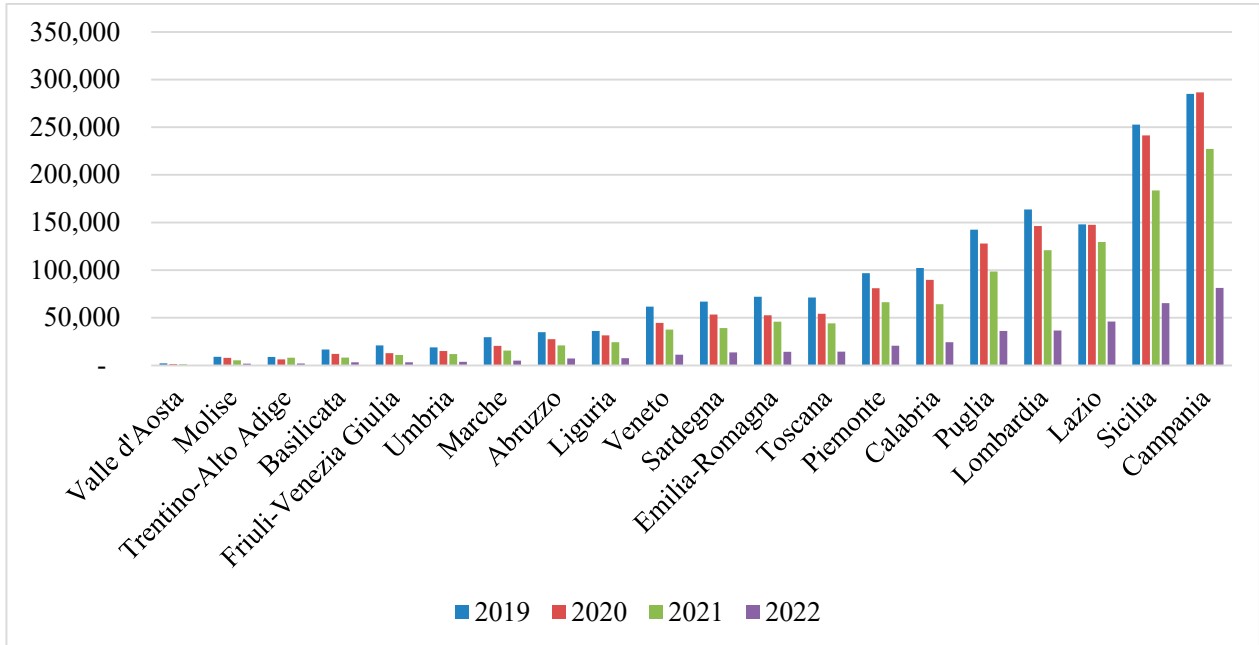

**Figure 12.** Citizenship Income household recipients by year and region, 2019–2022. Note: Households receiving at least one monthly Citizenship Income in the reference year by region. Source: own elaboration on the INPS database.

From 2019 to 2022, the average monthly amount increased by 50 euros, from 530 to 586 euros. The highest moderate monthly amounts are those in Campania (649 euros) and Sicily (629 euros). In northern Italy, the average monthly amount varies from 562 euros in Piedmont to 538 euros in Liguria (Figure 13).

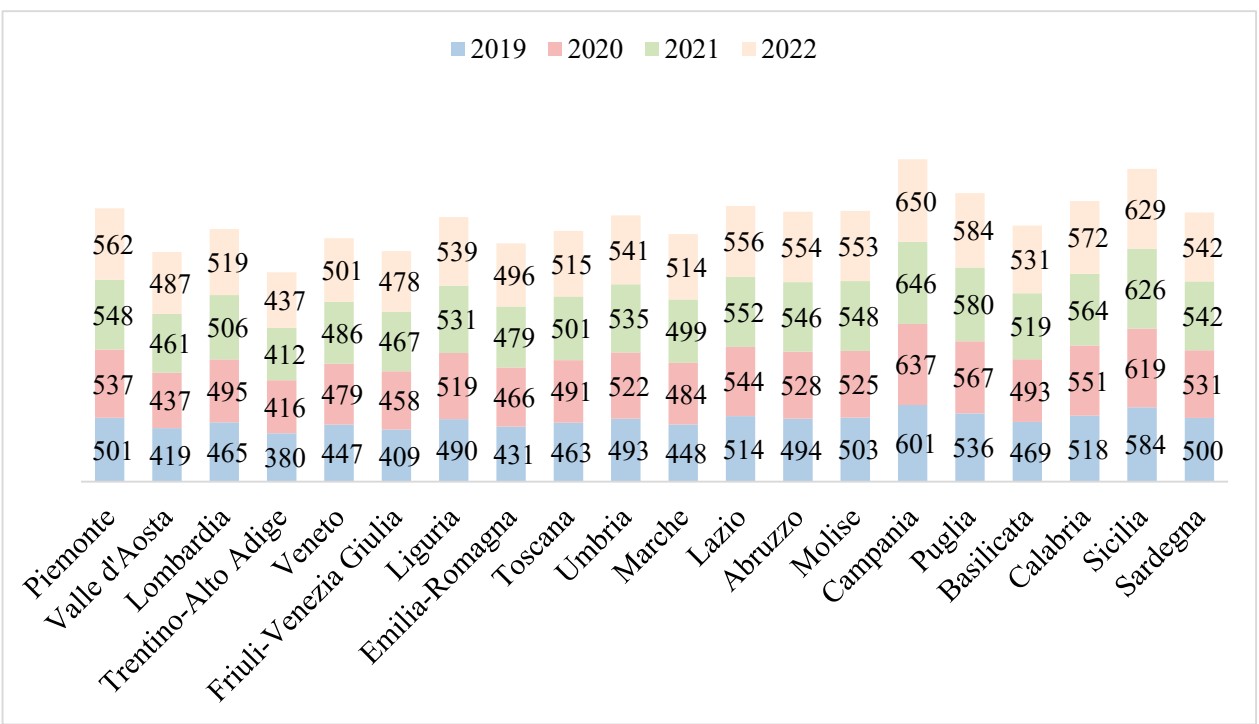

**Figure 13.** RdC per average monthly amount (in euros), by regions, 2019–2021. Source: own elaboration on the INPS database.

Concerning individuals' characteristics, in 2022, Italian citizens are the largest portion of beneficiaries (924,857 households, around 2 million individuals) whilst European citizens are the least represented (40,298 households and 86,556 individuals), alongside non-EU citizens (85,366 households and 228,594 individuals) (Figure 14). Non-national recipients are, in fact, underrepresented among beneficiaries. They represent, however, the largest quota of vulnerable people at the national level. This is due to the fact that one of the main requirements of the Citizenship Income is to be a resident in Italy for at least 10 years, thus excluding a relevant share of households in need.

Regarding the household composition of recipients, the highest proportion is represented by four-person households (477,508), followed by three-person households (353,736) and five-person households (285,285). The risk of poverty increases with the number of children in the household; nonetheless, the largest families are those with a limited rate of RdC recipients. As will be described in the next section (Section 4.4), the equivalence scale tends to disadvantage large families, thus failing to reach the target recipients of individuals more at risk of poverty (Figure 15).

In terms of equity, the policy measures efforts to reach the highest number of potential beneficiaries. Two main critical issues may be drawn from the above-mentioned data. The criteria for accessing the measure discourage access to large families with minors and foreign families. Another criticality concerns the assessment—for those who are qualified to receive the provision—of disposable resources (income, movable, and real estate wealth). Finally, regarding efficiency, the policy tool has limitations concerning the implementation of Employment and Social Inclusion Pacts. The latter are managed by social services at the local level. These elements are described in the following section (Section 4.4).

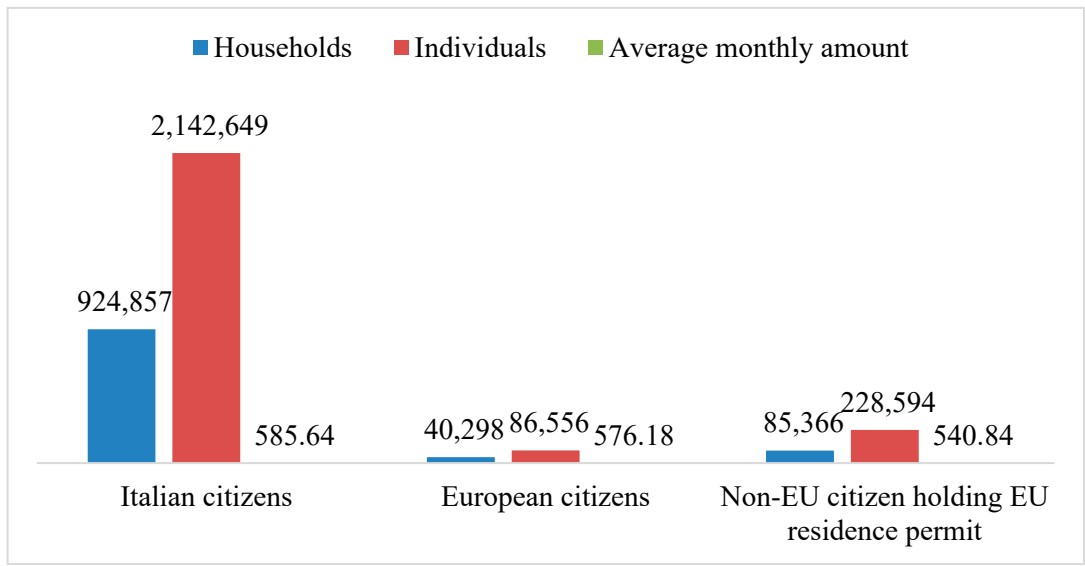

**Figure 14.** RdC per geographical origin, by household, individuals, and average monthly amount, 2022. Source: own elaboration on the INPS database.

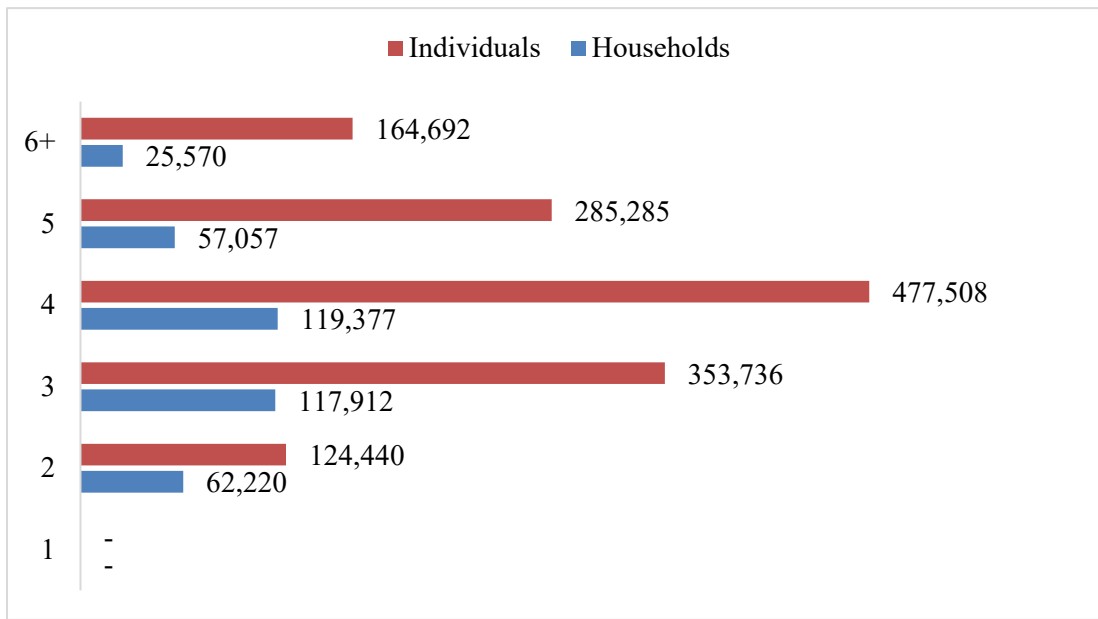

**Figure 15.** RdC recipients, by household composition, 2022. Source: own elaboration on the INPS database.

*4.4. The Measure's Constraints: The National Scientific Committee on Citizenship Income*

A comprehensive and coherent literature (Agostini et al. 2021; Baldini and Scarchilli 2021; Barbieri and Guarascio 2021; Natili et al. 2021; Vesan et al. 2021) has underlined the constraints of the Citizenship Income in combating poverty. In 2019, the Ministry of Labor and Social Policy appointed a Scientific Committee for the Assessment of the Citizenship Income, led by Professor Chiara Saraceno. In 2021, the Scientific Committee published a report on the actual functioning of Citizenship Income. In the meantime, the Anti-Poverty Network published its own proposal to reform the Italian minimum income.

Both organizations identified some ineffective dimensions in the design of the policy measure to be modified to make it more equitable and effective[15].

The critical points include (i) the eligibility criteria to access the measure; (ii) the gap concerning the amount of income support and the size of the family (including the members' age); (iii) the evaluation of the disposable resources (income, movable, and real estate wealth); (iv) the implementation of the Employment Pact; and (v) the Pact for Social Inclusion. As stated above, the first two points are mainly related to the equity dimension.

As for the first point, the eligibility criteria for accessing the measure foster a disadvantage for households with minors. Moreover, it excludes a large part of foreign families, including holders of international protection, and territorial (geographical) inequalities among recipients. The assessment of access requirements occurs in two stages. The first uses the ISEE criterion, which must not exceed 9360 euros per year. The second stage introduces three additional thresholds (independent of each other) for disposable income, movable wealth, and real estate. The equivalence scale assigns to each household type in the population a value in proportion to its needs. The factors used to assign these values are the size of the household and the age of its members, whether they are adults or children. The RdC equivalence scale allocates to the second and further components a coefficient of 0.4 if adults and 0.2 if children. The maximum coefficient, 2.1, increased to 2.2 for people with disabilities. The access threshold is thus more generous for families with adults. The households with minors—contrary to those composed of two or more adults—are usually penalized by the equivalence scale[16].

As far as non-Italian citizens are concerned, the applicant must be legally resident in Italy for at least 10 years. This excludes a large share of foreigners, including holders of international protection. The highest portion of foreigners live in northern Italy: here, the highest quota of poor, mainly, foreigners are excluded due to eligibility criteria.

As the second point, the income support addressed to families is higher if these are composed of adults[17]. It is the lowest for single-parent families (due to the equivalence scale) and it decreases with the presence of minor children. This results from the effect of the equivalence scale, as mentioned above, combined with the existence of a maximum amount. This second point is thus strictly connected to the first one. In relation to the geographical distribution, some estimates (Baldini et al. 2019) argue that the maximum ceiling, the amount of income support, should vary according to differences in the costs of living throughout Italy.

The third point concerns households' disposable resources (including movable and real estate wealth). The RdC amount is strictly dependent on the size of movable and real estate wealth; their value is considered as disposable income. Setting a strict maximum threshold for households' assets widened inequalities among potential recipients, who lost their right to support when they slightly exceeded the maximum threshold.

The last two points refer to the bottom-up implementation of the Pact for Employment and Social Exclusion. The implementation of both pacts reported strong temporal misalignment between the in-cash provision and their activation at the local level. The wide target of recipients has not been supported by a higher number of public servants employed in the social services. The activity slowed down during the pandemic due to restrictions and the emergency and widening inequalities in implementation spread throughout the whole country, fostering different modes of governance and coordination among actors at different levels of governance.

The unpreparedness—in terms of equity and efficiency—led governmental actors to introduce a temporary (contextual) measure: the Emergency Income. The policy tool was aimed at filling the gaps of the Citizenship Income, extending the degree of coverage to those excluded by the national minimum income. The implementation was released from the activation of the Pacts for Labor and Social Inclusion.

### 4.5. The Emergency Income: Filling the Gap but Only Temporarily

The Emergency Income was designed to fill the RdC gap, aiming to alleviate poverty across the country. The following data was retrieved from the INPS database, the one used for Citizenship Income. The largest quota of recipients resides in Lazio (79,042 households),

Campania (105,629), Lombardy (62,137), Sicily (96,783), and Puglia (50,445). Lombardy is one of the regions with the highest rate of beneficiaries, in contrast to the RdC (Figure 16).

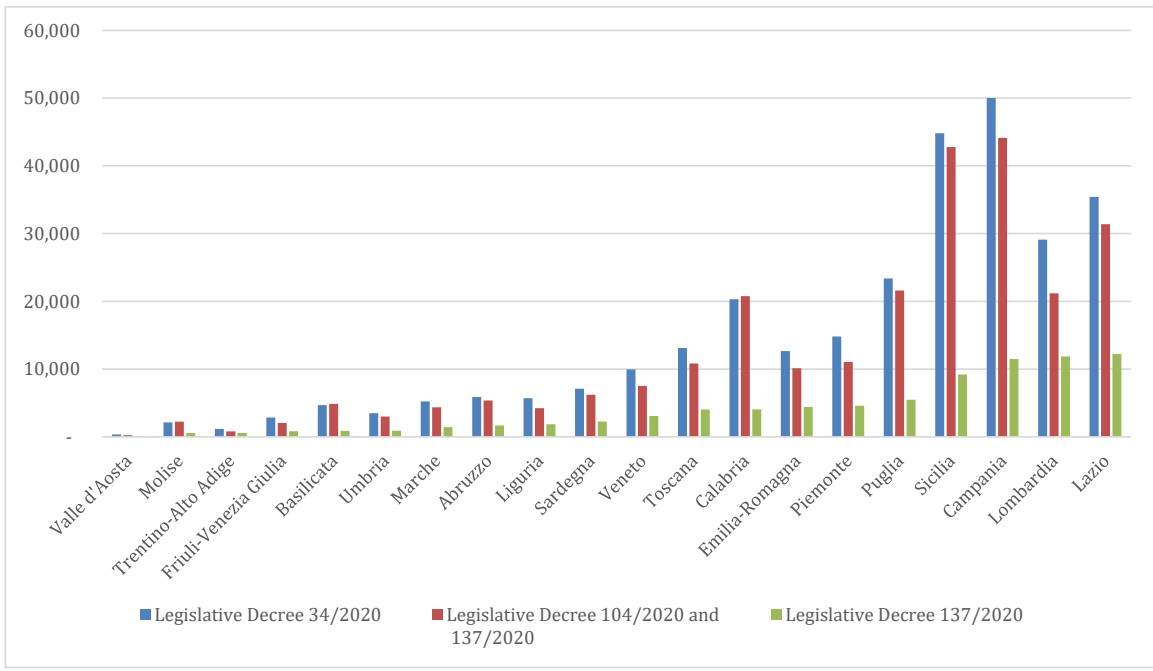

**Figure 16.** REM household recipients, by regions and Legislative Decree, 2020. Source: own elaboration on the INPS database.

In 2021, the largest portion of REM recipients were concentrated in Sicilia (170,427 households), Campania (160,970), Lazio (147,490), Lombardia (123,630), Puglia (104,340), and Calabria (91,919) (Figure 17).

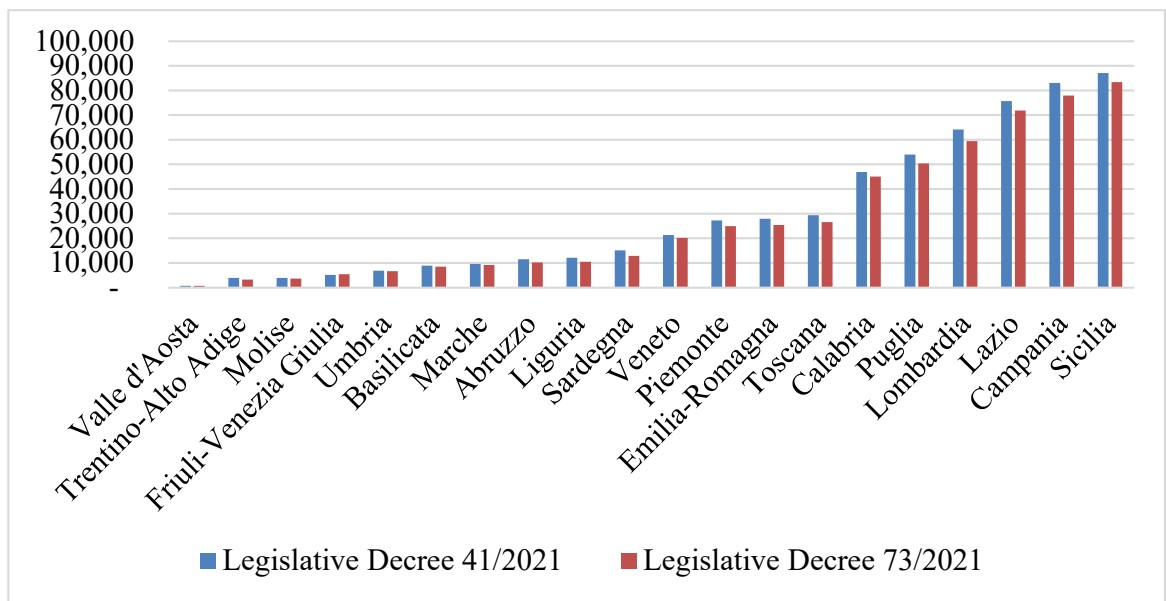

**Figure 17.** REM household recipients, by regions and Legislative Decree, 2021. Source: own elaboration on the INPS database.

As for the household composition, in 2021, four-person households (477,508) are over-represented among beneficiaries, followed by three-person households (353,736) and five-person ones (285,285) (Figure 18).

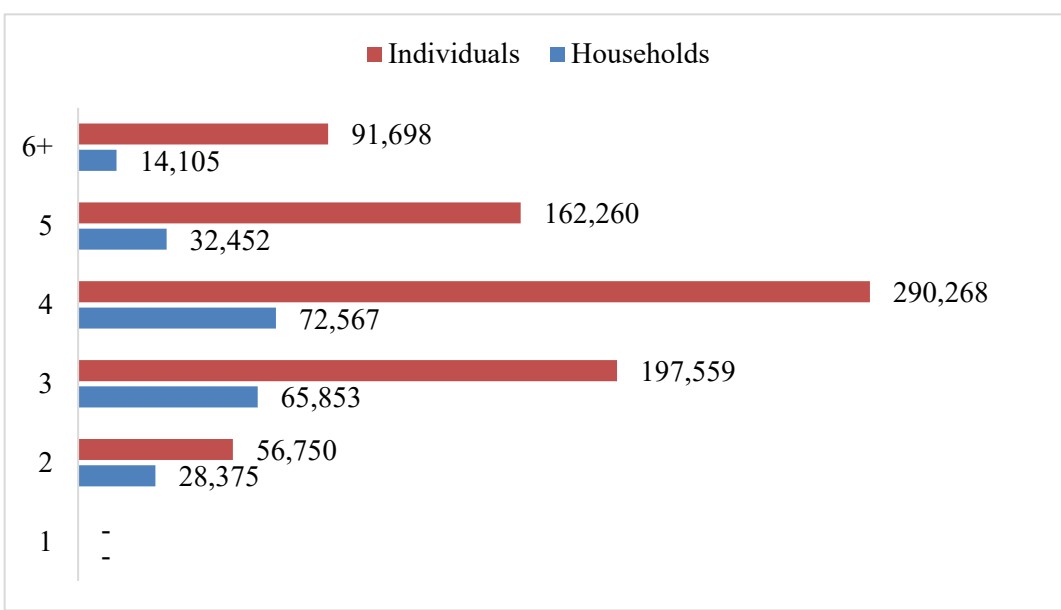

**Figure 18.** REM recipients, by household composition, 2021. Source: own elaboration on the INPS database.

The Emergency Income reached the population of youngsters, with more than one minor, tendentially larger than those covered by RdC. REM covered more households with employed workers and foreign citizens whilst the territorial distribution is mainly focused on the northern regions of Italy (INAPP 2020).

The REM succeeded in reaching households and individuals in hardship who manifested themselves before the pandemic. This also included families that, for several reasons, did not request the RdC, despite eligibility, and those excluded by welfare provisions due to their citizenship or movable/real estate assets. The REM covered those who fell into poverty during the pandemic because of restrictions (even loss) of working activities.

The adoption of complementary measures seems, initially, to make the case for readjusting the Citizenship Income towards more equitable distribution and effectiveness. As already said, the Scientific Committee and the Anti-Poverty Network formulated a set of proposals to strengthen the RdC effectiveness, widening the potential reachable audience through an easier mode of functioning. Nonetheless, the structural reform failed to come: the anti-poverty scheme was not reformed towards the above-mentioned goals.

## 5. Fostering Policy Change in Anti-Poverty Schemes: Still a Long Way to Go

In Italy, the adoption of anti-poverty schemes revealed a political complexity. On the one hand, broadly speaking, the guarantee of a basic income may be discouraged by the dynamics of political confrontation, within the core policy beliefs of the "left" and "right" parties (Jessoula and Madama 2015). On the other hand, in recent decades, inequality, poverty, and social exclusion have gained ground in public debate. This was due to socioeconomic hardship determined by technological, economic, and globalization transformations, including the 2008 financial and economic crisis. From 2010 onwards, policy-makers have been discussing ways to foster social policies, in the frame of EU institutions, in line with the Europe 2020 target of "lifting at least 20 million people out of the risk of poverty and social exclusion" (Curci et al. 2020).

Despite increasing attention and the pervasiveness of the phenomenon, Italy has long been lagging in the fight against poverty, showing strong path dependency. Poverty reduction policies have primarily been delegated to local governments, with nationwide programs geared mostly toward the elderly (e.g., pensions) and people with disabilities. However, the fragmentary social protection system became increasingly unable to protect a growing population of unemployed or precarious workers (Saraceno et al. 2020). From

2010 onwards, some relevant policy changes have occurred in Minimum Income schemes. The development of a large advocacy coalition (the "Alleanza contro la Povertà") managed to provide political protagonism to a heterogeneous social constituency, endowed with scarce political and instrumental resources, such as "poor individuals" are. It is worthwhile underlining that this achievement represented one of the most relevant changes influencing policy-making in the sector of anti-poverty policies in recent years (Gori 2020; Natili et al. 2021). The three streams—problem, policy, and politics—came together to launch the Italian minimum income scheme, the first anti-poverty scheme at the national level. However, this was not enough in terms of coverage capacity.

The COVID-19 pandemic proved the relevance of the Citizenship Income in combating poverty at the national level, also thanks to the new emergency measure introduced after the first lockdown in spring 2020. Poverty alleviation was possible with income support guaranteed by the two minimum income schemes operating between 2020 and 2021 during the climax of the crisis (Busso et al. 2021). If facilitating conditions to widen—and strengthen—the Citizenship Income seemed to be well-structured, a structural reform failed to occur in the Budget Law 2022.

The problem stream—poverty increasing—was recognized to be of public concern. The sharply increasing number of poor households brought the phenomenon to the top of the policy agenda. The absolute poverty rate went from 3.6% in 2005 to 7.7% in 2020, the highest rate ever reached. The COVID-19 pandemic shed light on the structural problems of the Italian welfare state: a high rate of unemployment, in-work poverty, educational poverty, and slow school-to-work transition. Facing an increasing poverty rate, the Citizenship Income failed to reach the widest audience of people in hardship. The measure's take-up rate was limited because of its eligibility criteria: large families with children and foreign people are the most disadvantaged. The measure, as shown in Section 4.3, tends to better address people living in the south of Italy, widening geographical inequalities among recipients. The problem stream converged, in the first half of 2020, with the policy stream: a new and temporary policy tool was introduced to cope with increasing poverty. The Emergency Income had weaker access criteria and aimed at supporting those who were ineligible for Citizenship Income. The measure was renewed three times during 2020 and 2021, thus demonstrating increasing political awareness of the need to widen the coverage of anti-poverty schemes.

The third stream for policy change, the political one, did not manage to meet the first two ones. The Budget Law 2022 introduced few marginal changes to Citizenship Income: the public resources were addressed to finance the control procedures related to set eligibility criteria. The Budget Law introduced three incremental changes. The national fund for RdC was increased annually by about 1.06 billion euros, thus reaching a total public expenditure of 8.4 billion euros for 2022. Tighter controls were envisaged to monitor the criminal records of family members and illegal transactions, facilitating data exchange between INPS and the Ministry of Justice. As for the Pact for Employment, the monthly payment ceases after refusing two job offers (previously it was three). The government aimed to strengthen workfare logics, reducing potential "disincentive effects" in the job search.

The COVID-19 pandemic played the role of a "focusing event" (Birkland 1998): it fostered the problem stream, with the increasing poverty rate and widening inequalities. *Why did it fail to achieve structural reform?* Poverty remains a divisive issue among political, institutional, and non-institutional stakeholders. The push toward introducing a minimum income was mostly to do with financial issues and sustainability constraints: improving the welfare state's financial coverage—thus, supporting vulnerable people to re-enter the labor market—alongside the need to strengthen redistributive mechanisms. However, its permanence in the policy agenda is still not secured. The rationales behind failed opportunities concern the political stream: the Citizenship Income is still a divisive issue among parties, coming continuously under attack by the two right-wing parties: Lega and Fratelli d'Italia. Moreover, there are different positions among institutional and non-

institutional stakeholders involved in policy decisions concerning the role of the RdC as an in-work benefit or not. Policy tools and strategies (provided by the Italian Anti-Poverty Network and the Scientific Committee) were in favor of policy changes. However, they were not "strong" enough to convince the Government to pursue a radical reform. At the end of 2021, much of the Government's attention and commitment were absorbed by the third COVID-19 wave of contagious disease and by the vaccination process (in turn, very divisive). There were no political conditions for fighting for the reform of the minimum income, only three years after its introduction.

**Author Contributions:** Conceptualization, F.M. and C.V.D.T.; methodology, F.M.; validation, F.M.; formal analysis, C.V.D.T.; investigation, C.V.D.T.; resources, F.M.; data curation, C.V.D.T.; writing—original draft preparation, F.M. and C.V.D.T.; writing—review and editing, F.M. and C.V.D.T.; supervision, F.M.; project administration, F.M. All authors have read and agreed to the published version of the manuscript.

**Funding:** This research received no external funding.

**Institutional Review Board Statement:** Not applicable.

**Informed Consent Statement:** Not applicable.

**Data Availability Statement:** Descriptive data about poverty are retrieved from ISTAT (The Italian National Statistics Institute (https://www.istat.it/it/, accessed on 25 June 2022) and Eurostat (https://ec.europa.eu/eurostat/data/database, accessed on 25 June 2022). Data about Citizenship and Emergency Income recipients are retrieved from National Social Insurance Agency (https://www.inps.it, accessed on 18 May 2022).

**Acknowledgments:** We acknowledge our anonymous referees for their revisions and feedback.

**Conflicts of Interest:** The authors declare no conflict of interest.

## Notes

[1]    According to Kingdon (1984), policy entrepreneurs may be elected politicians, leaders of interest groups, or unofficial spokespeople for particular causes.

[2]    The current paper is part of research work on poverty multidimensionality in Italy. More particularly, the authors have been involved in an Evaluation Study on the "New Policy Scenario", organized by Regione Lombardia and concerning anti-poverty policies in Lombardy (see Maino et al. 2021). Moreover, the authors contributed to the Deep Dive Report "Analysis of policies, programs, services, budgets, and mechanisms addressing Child Poverty and Social Exclusion in Italy" (see Unicef 2022). As a member of the Scientific Committee of the Alleanza contro la povertà, Franca Maino is currently involved in field research on the implementation mechanisms of the Citizenship Income.

[3]    The focus group involved seven Italian academics specializing in poverty and social exclusion in Italy. The focus group was held in May 2021.

[4]    The historical reconstruction is resumed by Agostini (2019); Gori (2020); Natili et al. (2021).

[5]    They concerned actions aimed at improving Active Labor Market Policies for adults' re-employment (active job search paths), and school performance and the protection of health among children and young people.

[6]    The "Inclusion Path" required the applicant's family unit to adhere to a personalized social and work activation project supported by an integrated network of interventions, identified by the social services of the municipalities and coordinated with Third Sector Organizations, social partners, and the whole local community. The activities may concern contacts with services, active job search, participation in training projects, school attendance and commitment, prevention, and health protection (for further information, see the Ministry of Labor and Social Policies website at https://www.lavoro.gov.it/temi-e-priorita/poverta-ed-esclusione-sociale/focus-on/Sostegno-per-inclusione-attiva-SIA/Pagine/default.aspx) (accessed on 18 May 2022).

[7]    The concept of "functional distortion" refers to the unequal distribution of protection of the various social risks (Ferrera 2012).

[8]    The "Sostegni-bis" Decree (DL 25 May 2021, no. 73) provided four months of REM for the months from June to September (art. 36), in addition to the three already arranged.

[9]    The Emergency Income was established by Article 82 of Decree-Law No. 34 of 19 May 2020. It was then renewed by three subsequent Decree Laws: Decree-Law No. 104 of 14 August (2020), No. 137 of 28 October (2020), and No. 73 of 25 May (2021).

[10]   The following data about poverty were retrieved from the Italian Statistical Institute open access database: https://www.istat.it (accessed on 25 June 2022).

[11]    According to ISTAT, absolute poverty defines households with a monthly expenditure equal to or less than the value of the absolute poverty threshold (which differs in size and composition by the age of the household, the geographical distribution, and the type of municipality of residence).

[12]    Inflation represents one of the major aspects of macroeconomic stability/instability: price jumps generally erode the real wages and assets of the poor more than those of the non-poor.

[13]    Italy is preceded by Spain (13.3%), Iceland (14.3%), and Romania (15.3%).

[14]    Further information about the Observatory is available here: https://www.inps.it/news/osservatorio-reddito-e-pensione-di-cittadinanza-i-dati-di-aprile (accessed on 18 May 2022 ).

[15]    Alongside the Scientific Committee for the Assessment of the Citizenship Income, the Italian Anti-Poverty Network launched eight proposals to reform the Citizenship Income. The proposals required: (1) making use of the ISEE equivalence scale (the criterion to evaluate the family economic situation), which would increase the number of families benefiting from the RdC by just under 400,000, thus extending access to households that are currently excluded due to the restrictive parameters chosen; (2) reducing the discriminatory 10-year residency restriction, bringing it back to the more reasonable 2-year level previously envisaged for the Inclusion Income (ReI); (3) loosening the additional constraint on movable assets, by raising the threshold to include those who are just above the margin, or making it more flexible; (4) supporting the individuals' and households' application, reintroducing the "One Stop Shop" (Unique Access Points) provided for the ReI; (5) developing a (personalized) take-over fostering integration between the Employment Centers (CpI) and local social services through the definition of joint work protocols, promoting the integrated use of the databases of the public entities involved in the implementation of the RdC (e.g., INPS, Municipalities, GEPI, MyAnpal); (6) promoting projects useful to the community (PUC) that are also useful to beneficiaries: making PUC volunteers according to a logic based on the empowerment and capacity building of the most vulnerable subjects; (7) providing well-functioning and targeted paths for updating and improving skills and a new design of the compatibility between the RdC and income from work, to avoid the poverty trap; and (8) promoting an employment-friendly Citizenship Income, reducing the marginal rate applied to earned income, lowering it from 100% to 60%.

[16]    Among the families with an ISEE below the threshold, those excluded from the RdC because the equivalent family income is higher than that established (the majority motivation in all cases) are 38% of the total, but they rise to 50% among those with minors, out of 27% of the remaining families. To a lesser extent, the same is true for movable and real estate assets, even if the latter plays a marginal role (Scientific Committee for the Assessment of the Citizenship Income 2021).

[17]    For whom the RdC reaches 26% of the median income of the same type of family.

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
