# Peer review of "Fostering Policy Change in Anti-Poverty Schemes in Italy: Still a Long Way to Go"

_socsci, doi:10.3390/socsci11080327_

Round 1

Reviewer 1 Report

The presentation of the Italian anti-poverty measures is interesting reading. It is also important to present the number of beneficiaries and how are they divided geographically to an international audience. However, in this present format, the manuscript is more like a research report than a scientific article. The content is very descriptive without a deeper analysis of the policies or the outcomes. According to the second sentence of the abstract, the article intends "to contribute to the discussion concerning policy window of opportunity and policy change, it underlines strong path dependency in conceptualization and implementation of Italy’s primary poverty measure, the Citizenship Income." Yet, the theoretical discussion on policy change is very marginal in the manuscript and the strong path dependency is not clearly shown in the argumentation. 

It seems that the manuscript is missing a coherent research design that is required for a scientific article. The content is informative but the article should have better elaborated and justified research questions/aims instead of exploring "three points about the poverty phenomenon and Italian anti-poverty measures" very broadly. A historical reconstruction of anti-poverty policies in Italy is interesting but it remains unclear, why the narrative contains four phases and why exactly these four periods with certain years can be defined as 'phases'. 

The manuscript is missing the conclusions that are derived from the presentation of the policy narrative or the empirical findings of the number of beneficiaries. It is confusing that the last paragraph of the manuscript opens up the 'big questions' regarding the welfare state that have not been discussed or introduced before. 

In addition, the manuscript contains unfinished text that includes too many language errors, unclear sentences with complicated structure, confusing content, or the mismatch between the text and the references, as in this sentence: “The Budget Law 2022 introduced, in fact, a few marginal changes (Lindblom 1979) to Citizenship Income concerning the public social expenditure addressed to finance the policy measure, the “control procedures” of eligibility criteria, and further peculiarity in policy design.” It should also be more clear what are the sources of information that are used in the argumentation. For example on page 4, there are findings on the poverty rates among the Italian workers without any references. ISTAT 2021 is used as a reference but not systematically. The INPS database should also be introduced. 

Author Response

The number of beneficiaries and how they are divided geographically have been added in section 4.3. We added analytical elements starting from the applied theoretical framework (§1) in our paper (multiple stream approach and policy change), contextualizing it in the Italian historical/political path towards the introduction of Citizenship Income, first, and Emergency Income, then. For this purpose, we re-structured our paper’s outline. Firstly, the research question and methodology are described (§2). The third section introduces the Citizenship and Emergency Income policy design (§3), including the main historical milestones that framed the introduction of the two measures. Secondly, the article analyses two ineffectiveness dimensions related to anti-poverty policies (§4): the increasing poverty rate in Italy, its take-up rate, including the characteristics of distribution among beneficiaries. The fifth section concludes.

To strengthen the analytical research section, we develop a dedicated section explaining the research questions, hypothesis, and methodology. The research question is “What are the main dimensions of the ineffectiveness of Citizenship Income? What are the rationales behind the failure of anti-poverty reform in 2022?”. The hypothesis states that poverty and growing inequality are still divisive themes among political and non-political actors (further explanations about hypotheses are presented in the research paper). In the framework of the Multiple Stream Approach and Policy Change literature, we tried to reframe our analysis and conclusions to underline how - considering peculiarities in the Italian welfare state - path dependency prevailed in achieving a structural reform. The two analytical dimensions of ineffectiveness to evaluate the policy measure - supported by data - are equity and efficiency.

Finally, the historical part was reduced and aimed at supporting analytical elements (equity and efficiency of RdC measure) throughout the paper and in the conclusion. Also, the conclusion has been reframed to include policy narratives presented in the introductive part (§1) and empirical findings described before (§4).

The used database and references have been adjusted according to referees’ requests and the journal requirements. The article has been professionally proofread. We will provide an official certificate to prove it.

Reviewer 2 Report

The contextualization with respect to previous and present theoretical background is too long, 8 from 18.

The application is very simple and unoriginal. It simply detects the type of individuals/ households receiving Citizenship Incomes. However, they do not propose anything new,  a new method or so on. In my opinion, it looks more like a simple report than a research work.

Author Response

(The authors gave the same response as above.)

Round 2

Reviewer 2 Report

In my opinion authors have done a very good work.

It is now ready for publication.